# Development of Seizures Following Traumatic Brain Injury: A Retrospective Study

**DOI:** 10.3390/jcm13185399

**Published:** 2024-09-12

**Authors:** Margaret Moran, Brooke Lajeunesse, Travis Kotzur, David Arian Momtaz, Daniel Li Smerin, Molly Frances Lafuente, Amirhossein Azari Jafari, Seyyedmohammadsadeq Mirmoeeni, Carlos Garcia, Paola Martinez, Kevin Chen, Ali Seifi

**Affiliations:** 1School of Medicine, University of Texas Health Science Center, San Antonio, TX 78229, USA; maggie.m.armstrong@gmail.com (M.M.); brookeaarrington@gmail.com (B.L.); kotzurt@livemail.uthscsa.edu (T.K.);; 2Department of Neurosurgery, University of Texas Health Science Center, San Antonio, TX 78229, USA; 3Department of Emergency Medicine, San Antonio Military Medical Center (SAMMC), San Antonio, TX 78234, USA; 4Student Research Committee, School of Medicine, Shahroud University of Medical Sciences, Shahroud 3614773955, Iran

**Keywords:** traumatic brain injury, post-traumatic seizure, seizures, subdural hematoma, mortality

## Abstract

**Objectives**: The multifaceted impact of Traumatic brain injury (TBI) encompasses complex healthcare costs and diverse health complications, including the emergence of Post-Traumatic Seizures (PTS). In this study, our goal was to discern and elucidate the incidence and risk factors implicated in the pathogenesis of PTS. We hypothesize that the development of PTS following TBI varies based on the type and severity of TBI. **Methods**: Our study leveraged the Nationwide Inpatient Sample (NIS) to review primary TBI cases spanning 2016–2020 in the United States. Admissions featuring the concurrent development of seizures during the admission were queried. The demographic variables, concomitant diagnoses, TBI subtypes, hospital charges, hospital length of stay (LOS), and mortality were analyzed. **Results**: The aggregate profile of TBI patients delineated a mean age of 61.75 (±23.8) years, a male preponderance (60%), and a predominantly White demographic (71%). Intriguingly, patients who encountered PTS showcased extended LOS (7.5 ± 9.99 vs. 6.87 ± 10.98 days, *p* < 0.001). Paradoxically, PTS exhibited a reduced overall in-hospital mortality (6% vs. 8.1%, *p* < 0.001). Notably, among various TBI subtypes, traumatic subdural hematoma (SDH) emerged as a predictive factor for heightened seizure development (OR 1.38 [1.32–1.43], *p* < 0.001). **Conclusions**: This rigorous investigation employing an extensive national database unveils a 4.95% incidence of PTS, with SDH accentuating odds of seizure risk by OR: 1.38 ([1.32–1.43], *p* < 0.001). The paradoxical correlation between lower mortality and PTS is expected to be multifactorial and necessitates further exploration. Early seizure prophylaxis, prompt monitoring, and equitable healthcare provision remain pivotal avenues for curbing seizure incidence and comprehending intricate mortality trends.

## 1. Introduction

Traumatic brain injury (TBI) is a prevailing and economically burdensome issue within the United States, exacting a substantial toll through heightened morbidity and mortality. The Centers for Disease Control and Prevention (CDC) has underscored the magnitude of this concern, projecting over 64,000 TBI-related fatalities in 2020 and more than 223,000 TBI-associated hospitalizations in 2019 alone [1,2]. The societal impact attributed to TBI is unequivocally reflected in the staggering estimated annual healthcare expenditure exceeding USD 40.6 billion [3]. Following TBI, patients are susceptible to various health impairments, psychosocial deficits, and neurological sequelae including the onset of post-traumatic seizures (PTS) [1,3,4]. Seizures occurring within one week following TBI are considered provoked secondary to head injury; this is considered early PTS. After one week, the development of multiple seizures is considered post-traumatic epilepsy and late PTS if only a single seizure occurs [5]. The development of post-traumatic seizures has been seen to occur in 2% to over 50% of patients following head trauma; the large range of PTS occurrence is attributed to injury severity and the type of TBI [6]. The development of PTS worsens the prognosis of TBI patients by increasing ICU/hospital length of stay and ICU ventilation requirements and results in poorer 24-month outcomes such as mortality and the further development of post-traumatic epilepsy [7]. PTE contributes to 3% to 6% of all new-onset epilepsy; therefore, further investigation into risk factors such as patient demographics and comorbidities, medical complications, and TBI subtype is warranted [8].

Early PTS occurrence is attributed to the interplay of primary injury and excitotoxic milieus. Subsequent PTS occurrence is linked to secondary injury-associated changes such as neuroinflammation and blood–brain barrier perturbations [4]. Pre-clinical models have shown that glutamate signaling and GABA-A channels could play a significant role in early PTS occurrence. It is hypothesized that increased glutamate signaling occurs in response to decreased GABAnergic activity, likely secondary to microRNA regulation [6]. In addition to glutamate and GABA-A, reactive gliosis secondary to the upregulation of c-Jun N-terminal kinase (JNK) and the activation of the JNK signaling pathway by glutamate have been shown to increase seizure activity following TBI in animal models [9]. A multitude of risk factors for early PTS have been reported, including factors such as advanced age, alcohol misuse, low-energy trauma, distinct intracranial injury patterns, pre-existing medical comorbidities, and TBI severity [10,11,12]. However, discrepancies persist between these risk factors and the findings within the current research landscape [7,13].

Comprehending and characterizing the incidence and risk factors for PTS are paramount, as they raise healthcare demands, increase economic burdens, and reduce functional outcomes [14,15]. In this study, we used a large national database and explored all patients with a principal admission for TBI. Research is scarce regarding the incidence of PTS following TBI on a national scale in the United States. Our intention in this review is to discern the incidence of in-hospital PTS and its associated risk factors. Our goal is to identify areas where preventive measures can be enacted to reduce PTS occurrence in the future. We hypothesize that the incident of seizure following TBI varies based on the type and severity of brain injury. 

## 2. Materials and Methods

The National Inpatient Sample (NIS) database from the Healthcare Cost and Utilization Project (HCUP) was queried for all primary cases with a principal diagnosis of traumatic brain injury from January 2016 to December 2020. NIS is the largest publicly available all-payer inpatient healthcare database in the United States, representing a 20% random stratified sample of all discharges from U.S. community hospitals, excluding rehabilitation and long-term acute care hospitals and yielding national estimates of hospital inpatient stays [16]. 

Using the International Classification of Diseases, Tenth Revision (ICD-10) codes, we extracted the ICD-10 code S06 family for intracranial injury (S06.0–S06.9) and ICD-10 code R56.1 for post-traumatic seizures. After properly excluding cases with incomplete and missing variables, the sample was divided into two groups based on the presence or absence of seizures secondary to TBI during the inpatient stay. Inclusion criteria included any patient with ICD-10 codes for TBI and post-traumatic seizures. Exclusion criteria included any patient with incomplete data.

Each group underwent analysis to compare demographics, including age, sex, race, income quartile level, insurance status, hospital length of stay (LOS), total hospital charges, adverse discharge (any discharge other than discharge to home), and in-hospital mortality. Multiple logistic regression models were created to elucidate relationships between seizure occurrence and the aforementioned variables, as well as the presence of deep vein thrombosis, sepsis, pneumonia, ventilator use, and other concurrent diagnoses. These regression models controlled for age, sex, ethnicity, and race when appropriate. 

Furthermore, the different types of TBI were analyzed to examine if a kind of injury had increased odds of causing a seizure. Each TBI type was categorized based on ICD-10 classification and analyzed independently through logistic regression models. 

### Statistical Analysis

Categorical results are reported as counts with column percentages. Continuous data are reported as means and standard deviations (SD); standard errors (SE) are provided where appropriate. A comparison of normally distributed data was performed with independent sample *t*-tests. For non-normally distributed data, the Wilcoxon rank-sum test was applied. Categorical variables were assessed with Fisher’s Exact Test or Chi-Square with Kendall Tau. Where appropriate, residuals were evaluated for normal distribution, and no multicollinearity was observed. Regression results are reported as an adjusted odds ratio (OR) with a 95% confidence interval (95% CI). For the analysis of days to the primary outcome (in-hospital mortality), Kaplan–Meier survival estimates and log-rank tests were used to compare the TBI with and without PTS. 

Statistical analyses were performed using R version 4.2.0 (R Core Team, 2022), a statistical computing environment developed by the R Foundation for Statistical Computing, Vienna, Austria. The significance level was set at *p* < 0.05. As the data extracted from the NIS are not linked to any patient identifiers, this study was deemed exempt from full IRB review by the University of Texas Health Science Center at San Antonio (HSC20150408N).

## 3. Results

### 3.1. Demographic Information

We reviewed 219,005 cases discharged for a principal diagnosis of TBI in the United States from 2016–2020 that were reported in the NIS database. Out of these, 10,838 patients developed seizures, which is a rate of 4.95%. Overall, TBI patients had a mean age of 61.75 (±23.8) years, were primarily male (60%), and mainly white (71%). Patients who developed seizures were significantly younger (mean age of 60.62 ± 20.41 years, *p* < 0.001), primarily male (62%, *p* < 0.001), and white (69%, *p* < 0.001). The majority of the cohort were insured by Medicare (53%); however, PTS patients were more likely to be insured by Medicaid (Medicaid insured 14% of patients without seizures vs. 19% with seizures, *p* < 0.001). Furthermore, the patients with PTS (1.30%) reported a significantly lower rate of severe TBI compared to patients without PTS (1.8%, *p* < 0.001). More information regarding the severity of injury, comorbidities, and other related demographic variables is shown in Table 1. Given the results in Figure 1, the incidence rate of PTS per 100,000 patients was 5115 in 2016, 4729 in 2017, 5094 in 2018, 4992 in 2019, and 4812 in 2020. Moreover, Table 2 presents the findings regarding the regression model analysis of the odds of having PTS following a primary TBI in the case of each demographic variable (Table 1 and Table 2; Figure 1).

### 3.2. Hospital Outcomes

Patients experiencing PTS had a longer hospital LOS (7.5 ± 9.99 vs. 6.87 ± 10.98 days, *p* < 0.001). Patients experiencing TBI with PTS had more adverse discharge dispositions than patients not experiencing PTS (47% vs. 45%, *p* < 0.001). There was a significant difference between the hospital charges between TBI without PTS vs. TBI with PTS (USD 98,237.05 (169,873.87) vs. USD 97,174.11 (135,125.81), *p* < 0.001) (Table 1).

### 3.3. Comorbid Conditions

As shown in Table 3, TBI patients who developed seizures were more likely to have hypertension (uncomplicated) (45% vs. 41%, *p* < 0.001), paralysis (9.3% vs. 5.3%, *p* < 0.001), liver disease (5.8% vs. 4%, *p* < 0.001), coagulopathy (12% vs. 10%, *p* < 0.001), fluid and electrolyte disorders (35% vs. 30%, *p* < 0.001), alcohol abuse (21% vs. 14%, *p* < 0.001), drug abuse (8% vs. 5.8%, *p* < 0.001), psychoses (3.3% vs. 1.4%, *p* < 0.001), and depression (16% vs. 12%, *p* < 0.001) (Table 3).

### 3.4. Comorbidity Predictive Factors

The regression analysis shown in Table 4 has demonstrated that comorbidities, such as psychoses (OR: 1.24, 95%CI [1.18–1.30], *p* < 0.001), AIDS/HIV (OR: 1.23, 95%CI [1.05–1.44], *p* = 0.011), alcohol abuse (OR: 1.15, 95%CI [1.12–1.17], *p* < 0.001), lymphoma (OR: 1.14, 95%CI [1.02–1.28], *p* = 0.026), solid tumor without metastasis (OR: 1.12, 95%CI [1.05–1.20], *p* = 0.001), and valvular disease (OR: 1.10, 95%CI [1.06–1.14], *p* < 0.001), were associated with significantly higher odds of having PTS following TBI. More details regarding the odds of developing PTS in terms of each comorbidity are shown in Table 4.

### 3.5. Type of TBI

Traumatic Subdural Hemorrhage (SDH) was the most common type of TBI in the whole cohort 105,554 (48%), as well as in the subgroup of PTS (57%) and in those without PTS (48%), *p* < 0.001, Table 1.

The regression analysis in Table 5 demonstrates the odds of having PTS following each subset of TBI. The only subtype of TBI that was a predictor of developing PTS was traumatic SDH (OR 1.38 [1.32–1.43], *p* < 0.001). However, traumatic cerebral edema (OR 0.52 [0.35–0.75], *p* < 0.001), concussion (OR: 0.68, 95%CI [0.62–0.74], *p* < 0.001), diffuse traumatic brain injury (OR: 0.74, 95%CI [0.65–0.84], *p* < 0.001), epidural hemorrhage (OR: 0.83, 95%CI [0.75–0.92], *p* < 0.001), and traumatic subarachnoid hemorrhage (OR: 0.80, 95%CI [0.76–0.83], *p* < 0.001) were associated with lower odds of developing PTS following TBI (Table 5).

### 3.6. Mortality and Survival

Given the fact that mortality was 8.0% in the entire TBI cohort, TBI patients without PTS had a higher mortality of 8.1% vs. 6% (Table 1).

When looking at additional data on hospice and day of death post-TBI, we found that TBI patients without PTS had a higher hospice rate: 7.7% hospice vs. 7.2%, *p* = 0.049. TBI patients without PTS died earlier during admission; they had a mean day of death post-TBI of 5.22 vs. 7.13 days, *p* < 0.001. Our Kaplan–Meier survival analysis demonstrated divergence in the survival fractions between TBI patients with and without PTS (log-rank *p* < 0.0001), Figure 2. Surprisingly, TBI patients who had associated PTS showed a higher survival rate (Figure 2).

### 3.7. Complications

In general, and as presented in Table 6, medical complications were found at significantly higher rates in TBI patients with PTS compared to those without PTS (47% vs. 44%, *p* < 0.001). Moreover, complications such as plegia and paresis (9.6% vs. 5.2%, *p* < 0.001) and stroke (2.4% vs. 1.6%, *p* < 0.001) were found at higher rates in the PTS group compared to patients without PTS.

Furthermore, the regression analysis in Table 7 demonstrates the odds of having a complication following the TBI if PTS was present. The highest odds were for stroke (OR 1.37 [1.2–1.56], *p* < 0.001), encephalopathy (OR 1.30 [1.23–1.37], *p* < 0.001), and plegia and paresis (OR 1.26 [1.19–1.34], *p* < 0.001) (Table 6 and Table 7).

## 4. Discussion

### 4.1. Incidence and Demographics

During the study period, 206,708 patients were identified as principal discharge diagnoses of TBI in the United States, with an in-hospital seizure rate of 4.95%. Increased PTS was statistically significant in TBI patients of younger age, male gender, white race, SDH, and those TBI patients who had comorbidities including stroke, uncomplicated hypertension, coagulopathy, alcohol abuse, drug abuse, paralysis, liver disease, psychoses, and depression. Ironically, PTS showed a protective association with mortality in this cohort. 

The rate of PTS in our study is similar to some previous studies [17]; however, we included all national TBI admissions, and we had a broader range of TBI severities as our cohort. Annegers, et al. reported an overall PTS incidence of 3.1% [16]. The incidence was 1.5% in patients with mild TBI, 2.9% for moderate TBI, and 17% for severe TBI [18]. A review of the available literature revealed a wide variability in reported incidence rates of PTS, ranging from 2% to 53.3%. The literature highlighted the influence of study population, methodology, and case definitions on these discrepancies. Despite the heterogeneity, similar to our study, other studies also contributed significant insights into the burden of PTS as a post-TBI complication, emphasizing the need for standardized criteria and further research to understand and manage this clinical phenomenon [5,19,20,21,22].

### 4.2. Traumatic Brain Injury Subtypes

The type of TBI had a statistically significant effect on PTS. Traumatic SDH was associated with 57% of cases, while epidural hemorrhage, traumatic subarachnoid hemorrhage, concussion, traumatic cerebral edema, and diffuse traumatic brain injury were associated with significantly fewer cases. The impact of traumatic SDH on PTS has been discussed previously in the literature. The epileptogenicity of SDH is secondary to the pronounced disruption of the blood–brain barrier (BBB), facilitating the influx of various inflammatory mediators into the brain parenchyma. This neuroinflammatory milieu lowers the seizure threshold by promoting neuronal hyperexcitability [23,24,25]. Furthermore, the extravasated blood and its breakdown products trigger a series of cytotoxic and neuroinflammatory responses, collectively contributing to an environment conducive to seizures via GABAnergic inhibitory neurotransmission perturbance, gliosis, and neuronal loss [26,27]. The increased incidence of epileptogenicity following SDH can be seen in several recent studies, with rates of 24% in acute SDH and 11% in chronic SDH [28,29,30]. On the contrary, some studies also report lower incidences of PTS following SDH, with one study hypothesizing that this may be secondary to the mechanism of injury that causes SDH, which may increase mortality, thereby decreasing reported rates of PTS [31].

### 4.3. Mortality

Our findings revealed an unexpected relationship between in-hospital PTS and mortality in patients with TBI. Contrary to conventional expectations, our results demonstrated that individuals with PTS exhibited a notably lower mortality rate. This counterintuitive finding challenges prevailing notions and warrants careful consideration. In our study, the mortality of PTS was 6.0%, compared to 8.1% in those without PTS. We hypothesized that this finding could be multifactorial. One possibility is more medical attention in PTS, more use of EEG, and better detection of seizures in recent years. Gilmore and colleagues discussed a similar counterintuitive connection in their investigation of PTS, wherein patients who experienced seizures demonstrated reduced mortality and favorable functional outcomes [32]. Similarly, a recent study on the Australian TBI registry showed that after adjustment for confounders, PTS was not associated with in-hospital mortality [7]. In a 23-year-long nationwide US study by Seifi et al., there was a reported decrease in mortality in traumatic SDH patients who developed hospital status epilepticus. This was attributed to increased detection secondary to more frequent EEG use in recent years [23,30]. A recent study utilizing the US national database revealed that the usage of EEG monitoring has been increasing by 11.2% annually in recent years [33]. This could support our study’s finding of lower mortality in TBI with PTS due to better detection. 

Another theory for this paradoxical finding is that those without reported PTS in the database had a more severe TBI and either died early on before having a chance to undergo EEG monitoring or possibly had more usage of hospice and palliative care due to the severity of the TBI. One other possibility is that although having continuous EEG monitoring can improve the outcomes in TBI, some hospitals do not have access to EEG and thus many TBI patients did not undergo EEG monitoring during admission. Therefore, mortality could happen before the healthcare team knew about the seizures [34,35]. 

TBI patients without PTS underwent hospice care more frequently than the subgroup of TBI with PTS (7.7% vs. 7.2%, *p* = 0.049). In addition, patients with TBI but without PTS died significantly earlier than the seizure with TBI subgroup (5.22 days vs. 7.13 days, *p* < 0.001). These findings also support our hypotheses for the paradoxical mortality, where patients with more severe TBI were hypothetically sicker, based on higher hospice rates and earlier days of death. Consequently, TBI patients who develop PTS receive more care, leading to a lower mortality rate. While this association demands further exploration, potential underlying mechanisms could involve heightened medical attention and intervention triggered by seizure episodes. This unexpected observation underscores the complexity of TBI outcomes and the need for comprehensive prospective research to unravel the intricate interplay between TBI, in-hospital seizures, and mortality.

### 4.4. Hospital Length of Stay (LOS)

Our study unveiled a significant and notable result: patients with PTS exhibited a substantially prolonged hospital LOS. This intriguing finding aligns with a study by Laing et al. which investigated the impact of post-traumatic seizures on hospitalization outcomes. In that study, patients who encountered PTS demonstrated a significantly extended hospital LOS and ICU LOS, emphasizing the potential influence of seizures on recovery trajectories and healthcare resource utilization [7]. Our discovery underscores the multifaceted nature of TBI-related complications and highlights the need for comprehensive management strategies that address the primary injury and its associated sequelae [35].

### 4.5. Race and Socioeconomics

Although in the primary analysis, the incidence of PTS was higher in the white race, in the regression analysis (Table 2), controlling for demographics, the black race had higher odds of PTS overall. Our results demonstrated that when adjusting for other variables, individuals of the black race exhibited higher PTS following TBI. This aligns with a growing body of literature highlighting health disparities in TBI outcomes among racial and socioeconomic groups. This finding resonates with a recent study based on the National Trauma Data Bank, which underscored the association between sociodemographic factors and post-TBI complications. The identified disparities emphasize the importance of addressing healthcare inequalities in TBI management and outcomes, ensuring equitable access to care for all population segments [36]. Our results suggest that patients in lower income quartiles and of lower socioeconomic status (qualifying for Medicaid) and with poor access to healthcare or lower health literacy (increased comorbidities) are more at risk of developing PTS. 

### 4.6. Complications

Our results (Table 6 and Table 7) showed that general medical complications, such as plegia and paresis and stroke, were significantly higher in TBI patients with PTS compared to those without PTS. In addition, the regression analysis revealed that stroke, encephalopathy, and plegia and paresis had the highest odds of developing following TBI in the presence of PTS. In this vein, studies have shown that stroke is one of the leading causes of seizures and epilepsy [37]. Moreover, severe strokes involving the cortex have the highest odds of developing post-stroke epilepsy. Furthermore, TBI is considered a risk factor for stroke and it is estimated that there is a 5.6% risk of epilepsy following TBI operated through intermediary stroke [37,38].

### 4.7. Limitations

Our study was queried from NIS, a national healthcare database. While large in number, making this study strong for the epidemiological understanding of PTS to design future prospective studies, this database had a few limitations. One major limitation is that the NIS database does not report clinical information such as the physical exams, laboratory work, imaging, and EEG findings of the patients, as well as medications, so we could not elaborate and use these critical variables in our study to better understand the patients who developed PTS. One main limitation is that the current incidence of PTS is based on whether the institutions used routine continuous EEG to detect PTS. However, some hospitals do not have continuous EEG or did not check for PTS. Another limitation of this study is the dependency of ICD-10 codes on the physicians’ notes and the coders’ judgment during the discharge process at the hospital. 

## 5. Conclusions

In summary, leveraging a large national database encompassing all TBI admissions across the United States, this comprehensive study has significantly contributed to understanding PTS and associated risk factors. With an observed incidence rate of 4.95% for seizures following diverse types and severities of TBI, this research underscores the substantial risk TBI poses in precipitating seizures. SDH emerged as a particularly predisposing factor, elevating seizure risk by up to 60%. Considering this high association, further research is warranted regarding early seizure prophylaxis and screening via techniques such as EEG in TBI management. Addressing healthcare disparities, particularly in marginalized populations, and enhancing health literacy to manage comorbidities effectively is a pivotal strategy to diminish PTS incidence. Surprisingly, the counterintuitive finding of lower mortality among PTS patients warrants further investigation to elucidate the underlying mechanisms. Ultimately, identifying risk factors for PTS serves as a foundational step toward implementing timely preventative measures, which not only holds the potential to curtail mortality rates but also to alleviate the financial burden and mitigate adverse health outcomes associated with PTS. Further prospective research will undoubtedly shed light on the complex interplay between TBI, seizures, and mortality, guiding the development of tailored interventions to optimize patient outcomes and healthcare resource utilization.

## Figures and Tables

**Figure 1 jcm-13-05399-f001:**
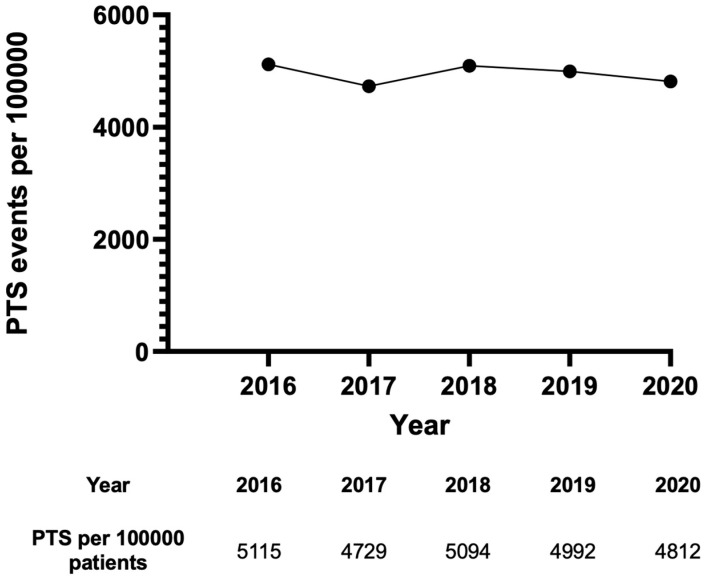
The incidence of PTS per 100,000 patients from 2016–2020.

**Figure 2 jcm-13-05399-f002:**
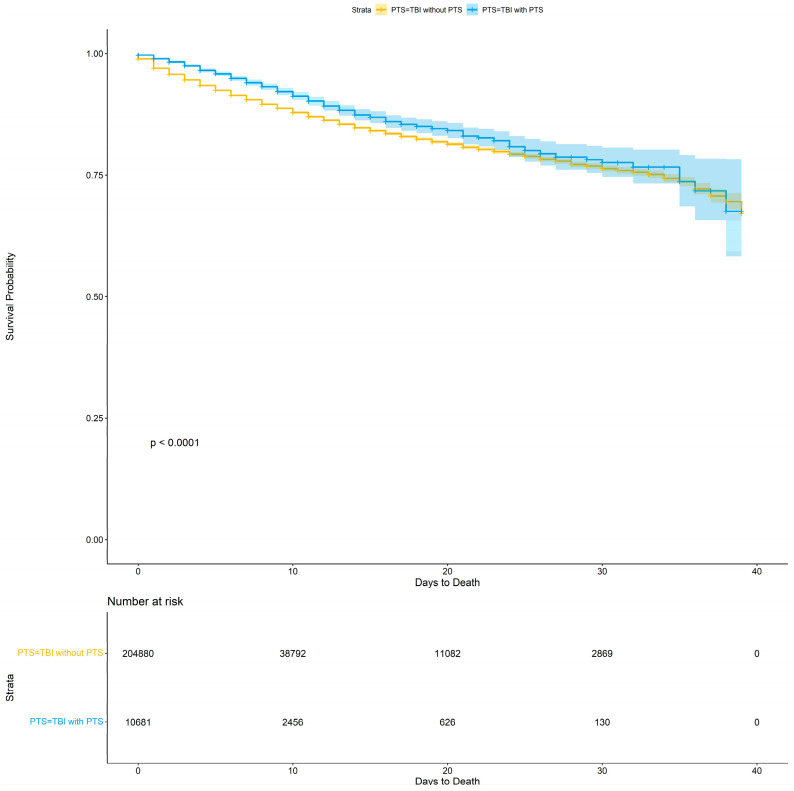
Kaplan–Meier survival analysis of TBI patients with and without seizures.

**Table 1 jcm-13-05399-t001:** Demographic information and hospital outcomes of included TBI patients with and without post-traumatic seizures.

Demographics	Overall, N = 219,005 ^1^	TBI w/o PTS, N = 208,167 ^1^	TBI w/PTS, N = 10,838 ^1^	*p*-Value ^2^
Age	61.75 (23.80)	61.81 (23.96)	60.62 (20.41)	<0.001
Age Category				<0.001
<55	69,400 (32%)	65,819 (32%)	3581 (33%)	
55–74	64,585 (29%)	60,472 (29%)	4113 (38%)	
75–89	67,889 (31%)	65,132 (31%)	2757 (25%)	
90+	17,116 (7.8%)	16,729 (8.0%)	387 (3.6%)	
Gender				<0.001
Female	88,178 (40%)	84,064 (40%)	4114 (38%)	
Male	130,764 (60%)	124,040 (60%)	6724 (62%)	
Race/Ethnicity				<0.001
Asian or Pacific Islander	7631 (3.5%)	7386 (3.5%)	245 (2.3%)	
Black	22,512 (10%)	20,978 (10%)	1534 (14%)	
Hispanic	24,818 (11%)	23,699 (11%)	1119 (10%)	
Native American	1750 (0.8%)	1611 (0.8%)	139 (1.3%)	
Other	7775 (3.6%)	7437 (3.6%)	338 (3.1%)	
White	154,519 (71%)	147,056 (71%)	7463 (69%)	
Household income by zip code				<0.001
0–25th percentile	60,418 (28%)	57,175 (27%)	3243 (30%)	
26th to 50th percentile	56,371 (26%)	53,617 (26%)	2754 (25%)	
51st to 75th percentile	53,778 (25%)	51,133 (25%)	2645 (24%)	
76th to 100th percentile	48,438 (22%)	46,242 (22%)	2196 (20%)	
Rural/Urban				0.2
Rural	13,916 (6.4%)	13,259 (6.4%)	657 (6.1%)	
Urban	205,089 (94%)	194,908 (94%)	10,181 (94%)	
Payer				<0.001
Medicaid	30,938 (14%)	28,827 (14%)	2111 (19%)	
Medicare	115,646 (53%)	109,718 (53%)	5928 (55%)	
Other	11,243 (5.1%)	10,778 (5.2%)	465 (4.3%)	
Private insurance	49,247 (22%)	47,433 (23%)	1814 (17%)	
Self-pay	11,931 (5.4%)	11,411 (5.5%)	520 (4.8%)	
Time to Any First Procedure	1.02 (3.05)	1.01 (3.05)	1.10 (3.04)	<0.001
Length of Stay	6.90 (10.94)	6.87 (10.98)	7.50 (9.99)	<0.001
Total Charge	98,184.45 (168,322.98)	98,237.05 (169,873.87)	97,174.11 (135,125.81)	<0.001
Discharge Disposition				<0.001
Adverse Discharge	97,828 (45%)	92,784 (45%)	5044 (47%)	
Home Discharge with Care	29,404 (13%)	27,904 (13%)	1500 (14%)	
Routine Discharge	91,645 (42%)	87,357 (42%)	4288 (40%)	
Hospice	16,735 (7.6%)	15,960 (7.7%)	775 (7.2%)	0.049
Severity of Injury				<0.001
Extreme Loss of Function	35,971 (16%)	34,255 (16%)	1716 (16%)	
Major Loss of Function	59,142 (27%)	55,901 (27%)	3241 (30%)	
Minor Loss of Function	43,397 (20%)	41,804 (20%)	1593 (15%)	
Moderate Loss of Function	80,479 (37%)	76,191 (37%)	4288 (40%)	
Severe TBI	3792 (1.7%)	3651 (1.8%)	141 (1.3%)	<0.001
Mortality	17,509 (8.0%)	16,862 (8.1%)	647 (6.0%)	<0.001
Day of Death	5.29 (8.90)	5.22 (8.96)	7.13 (7.13)	<0.001
Elixhauser Comorbidity Index				<0.001
≤0	72,114 (33%)	71,768 (34%)	346 (3.2%)	
1–2	6915 (3.2%)	6726 (3.2%)	189 (1.7%)	
3–4	12,452 (5.7%)	11,801 (5.7%)	651 (6.0%)	
5+	127,524 (58%)	117,872 (57%)	9652 (89%)	
Concussion	16,401 (7.5%)	15,904 (7.6%)	497 (4.6%)	<0.001
Traumatic Cerebral Edema	882 (0.4%)	856 (0.4%)	26 (0.2%)	0.008
Diffuse Traumatic Brain Injury	5919 (2.7%)	5693 (2.7%)	226 (2.1%)	<0.001
Focal Traumatic Brain Injury	18,369 (8.4%)	17,481 (8.4%)	888 (8.2%)	0.2
Epidural Hemorrhage	8786 (4.0%)	8417 (4.0%)	369 (3.4%)	<0.001
Traumatic Subdural hemorrhage	105,554 (48%)	99,357 (48%)	6197 (57%)	<0.001
Traumatic Subarachnoid Hemorrhage	59,157 (27%)	56,731 (27%)	2426 (22%)	<0.001
Other Specified Intracranial Injuries	1155 (0.5%)	1089 (0.5%)	66 (0.6%)	0.5
Unspecified Intracranial Injury	6603 (3.0%)	6278 (3.0%)	325 (3.0%)	>0.9

^1^ Mean (SD); n (%). ^2^ Wilcoxon rank-sum test; Pearson’s Chi-squared test.

**Table 3 jcm-13-05399-t003:** Comorbidities related to included Traumatic Brain Injury (TBI) patients with and without post-traumatic seizures (PTS).

Comorbidity	Overall, N = 219,005 ^1^	TBI w/o PTS, N = 208,167 ^1^	TBI w/PTS, N = 10,838 ^1^	*p*-Value ^2^
Congestive heart failure	24,812 (11%)	23,799 (11%)	1013 (9.3%)	<0.001
Cardiac arrhythmias	56,031 (26%)	53,478 (26%)	2553 (24%)	<0.001
Valvular disease	11,704 (5.3%)	11,142 (5.4%)	562 (5.2%)	0.5
Pulmonary circulation disorders	5245 (2.4%)	5014 (2.4%)	231 (2.1%)	0.066
Peripheral vascular disorders	12,192 (5.6%)	11,622 (5.6%)	570 (5.3%)	0.2
Hypertension (uncomplicated)	89,900 (41%)	85,006 (41%)	4894 (45%)	<0.001
Hypertension (complicated)	35,248 (16%)	33,710 (16%)	1538 (14%)	<0.001
Paralysis	12,028 (5.5%)	11,015 (5.3%)	1013 (9.3%)	<0.001
Other neurologic disorders	54,002 (25%)	43,164 (21%)	10,838 (100%)	<0.001
Chronic pulmonary disease	27,800 (13%)	26,322 (13%)	1478 (14%)	0.002
Diabetes (uncomplicated)	23,391 (11%)	22,285 (11%)	1106 (10%)	0.10
Diabetes (complicated)	24,469 (11%)	23,302 (11%)	1167 (11%)	0.2
Hypothyroidism	27,147 (12%)	25,752 (12%)	1395 (13%)	0.12
Renal failure	26,282 (12%)	25,166 (12%)	1116 (10%)	<0.001
Liver disease	9046 (4.1%)	8422 (4.0%)	624 (5.8%)	<0.001
Peptic ulcer disease (excluding bleeding)	757 (0.3%)	723 (0.3%)	34 (0.3%)	0.6
AIDS/HIV	386 (0.2%)	353 (0.2%)	33 (0.3%)	0.001
Lymphoma	1254 (0.6%)	1194 (0.6%)	60 (0.6%)	0.8
Metastatic cancer	2571 (1.2%)	2457 (1.2%)	114 (1.1%)	0.2
Solid tumor without metastasis	3071 (1.4%)	2901 (1.4%)	170 (1.6%)	0.13
Rheumatoid arthritis/collagen vascular diseases	4497 (2.1%)	4274 (2.1%)	223 (2.1%)	>0.9
Coagulopathy	22,539 (10%)	21,243 (10%)	1296 (12%)	<0.001
Obesity	12,162 (5.6%)	11,588 (5.6%)	574 (5.3%)	0.2
Weight loss	15,293 (7.0%)	14,549 (7.0%)	744 (6.9%)	0.6
Fluid and electrolyte disorders	66,947 (31%)	63,172 (30%)	3775 (35%)	<0.001
Blood loss anemia	1137 (0.5%)	1073 (0.5%)	64 (0.6%)	0.3
Deficiency anemias	5292 (2.4%)	4951 (2.4%)	341 (3.1%)	<0.001
Alcohol abuse	32,401 (15%)	30,083 (14%)	2318 (21%)	<0.001
Drug abuse	12,947 (5.9%)	12,083 (5.8%)	864 (8.0%)	<0.001
Psychoses	3284 (1.5%)	2930 (1.4%)	354 (3.3%)	<0.001
Depression	27,610 (13%)	25,838 (12%)	1772 (16%)	<0.001

^1^ n (%). ^2^ Pearson’s Chi-squared test.

**Table 4 jcm-13-05399-t004:** Binomial regression analysis of the odds of developing post-traumatic seizures (PTS) following any primary traumatic brain injury (TBI), controlling for age, gender, race, and injury severity. Odds ratios (OR) with 95% confidence intervals (CI) are presented, with statistical significance indicated at *p* < 0.05.

Comorbidity	Odds Ratio	Lower 95% CI	Upper 95% CI	*p*-Value
Congestive heart failure	0.89	0.85	0.92	<0.001
Cardiac arrhythmias	0.97	0.95	0.99	0.009
Valvular disease	1.10	1.06	1.14	<0.001
Pulmonary circulation disorders	1.01	0.95	1.07	0.796
Peripheral vascular disorders	1.04	1.00	1.08	0.051
Hypertension (uncomplicated)	0.99	0.97	1.01	0.491
Hypertension (complicated)	1.04	0.99	1.09	0.13
Paralysis	1.01	0.98	1.04	0.55
Chronic pulmonary disease	1.03	1.00	1.05	0.037
Diabetes (uncomplicated)	0.94	0.91	0.97	<0.001
Diabetes (complicated)	0.94	0.91	0.97	<0.001
Hypothyroidism	1.06	1.03	1.09	<0.001
Renal failure	0.91	0.87	0.95	<0.001
Liver disease	0.99	0.95	1.03	0.599
Peptic ulcer disease (excluding bleeding)	0.82	0.70	0.95	0.008
AIDS/HIV	1.23	1.05	1.44	0.011
Lymphoma	1.14	1.02	1.28	0.026
Metastatic cancer	0.96	0.89	1.05	0.376
Solid tumor without metastasis	1.12	1.05	1.20	0.001
Rheumatoid arthritis	0.99	0.93	1.05	0.702
Coagulopathy	1.04	1.01	1.07	0.005
Obesity	0.94	0.91	0.98	0.002
Weight loss	0.83	0.80	0.86	<0.001
Fluid and electrolyte disorders	0.92	0.90	0.94	<0.001
Blood loss anemia	1.08	0.97	1.21	0.159
Deficiency anemias	1.02	0.97	1.07	0.544
Alcohol abuse	1.15	1.12	1.17	<0.001
Drug abuse	1.00	0.96	1.03	0.791
Psychoses	1.24	1.18	1.30	<0.001
Depression	1.00	0.98	1.03	0.769

**Table 5 jcm-13-05399-t005:** Binomial regression analysis of the odds of developing post-traumatic seizures (PTS) following different subsets of traumatic brain injury (TBI), adjusting for age, race, gender, and injury severity, as well as comorbidity burden using the Elixhauser Comorbidity Index. Odds ratios (OR) with 95% confidence intervals (CI) are presented, with statistical significance indicated at *p* < 0.05.

Type of TBI	Odds Ratio	Lower 95% CI	Upper 95% CI	*p*-Value
Concussion	0.68	0.62	0.74	<0.001
Traumatic Cerebral Edema	0.52	0.35	0.75	<0.001
Diffuse Traumatic Brain Injury	–0.74	0.65	0.84	<0.001
Focal Traumatic Brain Injury	0.97	0.90	1.03	0.319
Epidural Hemorrhage	0.83	0.75	0.92	<0.001
Traumatic Subdural Hemorrhage	1.38	1.32	1.43	<0.001
Traumatic Subarachnoid Hemorrhage	0.80	0.76	0.83	<0.001
Other Specified Intracranial Injuries	1.16	0.91	1.47	0.225
Unspecified Intracranial Injury	1.05	0.94	1.17	0.386

**Table 6 jcm-13-05399-t006:** Complications in TBI patients for both groups (with and without PTS).

Complication	Overall, N = 219,005 ^1^	TBI w/o PTS, N = 208,167 ^1^	TBI w/PTS, N = 10,838 ^1^	*p*-Value ^2^
Medical Complications	96,110 (44%)	91,050 (44%)	5060 (47%)	<0.001
Respiratory Failure	35,501 (16%)	33,837 (16%)	1664 (15%)	0.013
Pulmonary Embolism	1367 (0.6%)	1304 (0.6%)	63 (0.6%)	0.6
Pneumonia	10,442 (4.8%)	10,049 (4.8%)	393 (3.6%)	<0.001
Cardiac Arrest	2904 (1.3%)	2814 (1.4%)	90 (0.8%)	<0.001
Heart Failure	21,769 (9.9%)	20,896 (10%)	873 (8.1%)	<0.001
Myocardial Infarction	2397 (1.1%)	2296 (1.1%)	101 (0.9%)	0.10
Transfusion	7531 (3.4%)	7254 (3.5%)	277 (2.6%)	<0.001
Deep Vein Thrombosis	3844 (1.8%)	3628 (1.7%)	216 (2.0%)	0.053
Acute Kidney Disease	20,336 (9.3%)	19,495 (9.4%)	841 (7.8%)	<0.001
Urological Infections	19,227 (8.8%)	18,232 (8.8%)	995 (9.2%)	0.13
Stroke	3609 (1.6%)	3354 (1.6%)	255 (2.4%)	<0.001
Delirium	4739 (2.2%)	4501 (2.2%)	238 (2.2%)	0.8
Plegia and Paresis	11,879 (5.4%)	10,834 (5.2%)	1045 (9.6%)	<0.001
Osteomyelitis	300 (0.1%)	286 (0.1%)	14 (0.1%)	0.8
Sepsis	5637 (2.6%)	5400 (2.6%)	237 (2.2%)	0.009

^1^ n (%); ^2^ Pearson’s Chi-squared test.

**Table 7 jcm-13-05399-t007:** Binomial regression analysis of the odds of experiencing complications following traumatic brain injury (TBI) in the presence of post-traumatic seizures (PTS). The model adjusts for potential confounders, including gender, age, race, socioeconomic status, injury severity, and comorbidity burden as measured by the Elixhauser Comorbidity Index. Odds ratios (OR) are reported with 95% confidence intervals (CI), and statistical significance was determined at a *p*-value threshold of <0.05.

Complication	Odds Ratio	Lower 95% CI	Upper 95% CI	*p*-Value
Medical Complication	0.94	0.92	0.96	<0.001
Respiratory Failure	0.95	0.91	1.00	0.063
Pulmonary Embolism	0.81	0.65	1.01	0.057
Pneumonia	0.71	0.64	0.79	<0.001
Cardiac Arrest	0.62	0.50	0.77	<0.001
Heart Failure	0.64	0.59	0.69	<0.001
Myocardial Infarction	0.81	0.66	1.00	0.046
Deep Vein Thrombosis	1.04	0.90	1.19	0.628
Acute Kidney Disease	0.73	0.69	0.78	<0.001
Urological Infection	1.03	0.97	1.09	0.407
Stroke	1.37	1.20	1.56	<0.001
Delirium	1.05	0.93	1.19	0.438
Encephalopathy	1.30	1.23	1.37	<0.001
Plegia and paresis	1.26	1.19	1.34	<0.001
Osteomyelitis	0.77	0.47	1.26	0.295
Sepsis	0.78	0.68	0.89	<0.001

**Table 2 jcm-13-05399-t002:** Logistic regression analysis of the odds of developing post-traumatic seizures (PTS) following a primary traumatic brain injury (TBI), stratified by demographic variables. The model includes adjustments for age, gender, race, and comorbidity burden as measured by the Elixhauser Comorbidity Index. Odds ratios (OR) with 95% confidence intervals (CI) are presented, with statistical significance indicated at *p* < 0.05.

Demographic Variables	Odds Ratio	Lower 95% CI	Upper 95% CI	*p*-Value
Age < 50	1.00			
Age 55–74	0.83	0.79	0.87	<0.001
Age 75–89	0.45	0.42	0.47	<0.001
Age 90+	0.24	0.22	0.27	<0.001
Female	1.01	0.97	1.05	0.568
White	1.00			
Black	1.29	1.22	1.36	<0.001
Hispanic	0.93	0.88	0.99	0.033
Asian or Pacific Islander	0.71	0.63	0.80	<0.001
Native American	1.40	1.18	1.66	<0.001
Other	0.89	0.80	1.00	0.04
Elixhauser ≤ 0	1.00			
Elixhauser 1–2	7.67	6.46	9.10	<0.001
Elixhauser 3–4	13.68	12.05	15.52	<0.001
Elixhauser 5+	24.98	22.47	27.76	<0.001
Minor Loss of Function	1.00			
Moderate Loss of Function	0.76	0.72	0.81	<0.001
Major Loss of Function	0.60	0.56	0.63	<0.001
Extreme Loss of Function	0.41	0.38	0.44	<0.001

^1^ n (%). ^2^ Pearson’s Chi-squared test.

## Data Availability

The HCUP raw data is available to researchers only upon purchase from the government and it is not allowed to be published publicly.

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
