# Peer review of "Development of Seizures Following Traumatic Brain Injury: A Retrospective Study"

_jcm, 2024, doi:10.3390/jcm13185399_

Round 1

Reviewer 1 Report

Comments and Suggestions for Authors

This is a study using the NIS database to examine the incidence of early onset seizures after Traumatic brain injury.  The study is very detailed and accurate for what it examines but is limited by the data available to them for the analysis.  Out of the 219,000 cases examined and 10,800 with seizures during the admission this gives great statistical power for the population enrolled in the study.

The data shows a population skewed to an older age (mean = 61.75 yr) this is not the population regularly seen with TBI in the population - I.e. 15 - 25 years.  The male to female ratio of 60:40 male to female is close to the usual reporting of 67%;33% but the demographic data clearly indicates that the population on were largely white 71%, urban residents 94% 

Of interest is the Comorbidity index at the 5+ level and the high rate of post traumatic seizures.   There also is as suspected a correlation to the pathophysiology of the TBI with SDH having the greatest correlation with the PTS.

It was not too surprising that those without PTS had a higher mortality rate.  Long term survival I s also not correlated to seizures or Post traumatic epilepsy.

The discussion comparing their findings to other studies as being similar is reassuring and sound and the discussion of pathophysiology of SDH and Seizures seems sound but the discussions in the following paragraphs  line 240 - 256 are interesting but not needed to discuss this study and their findings.

The review of their limitations with the data from the NIS is helpful but the lack of EEG data is disconcerting, especially when it is now common practice for acute continuous EEG monitoring for those in ICU, at the highest risk of PTS.  Also noticeable is the lack of information regarding the use of prophylactic anticonvulsants in the first 7 days post TBI or what forms of neurosurgical intervention were needed or performed in response to the intracranial pathology which may be a key marker of severity/ complexity of the trauma.

Finally their conclusions/recommendations over describe the NIS database as Robust - it is large in numbers but limited in other clinical ways and its population sample.  They also purport that early prophylaxis or AED use will lead to reduced PTS incidence ( a conclusion that may be correct but they have not proven) and I believe it is clear in the literature that distinction between early onset seizures and Late onset seizure post TBI need to be distinguished and addressed clinically in different ways.

Author Response

Response to Reviewer 1:

This is a study using the NIS database to examine the incidence of early onset seizures after Traumatic brain injury.  The study is very detailed and accurate for what it examines but is limited by the data available to them for the analysis.  Out of the 219,000 cases examined and 10,800 with seizures during the admission this gives great statistical power for the population enrolled in the study.

The data shows a population skewed to an older age (mean = 61.75 yr) this is not the population regularly seen with TBI in the population - I.e. 15 - 25 years.  The male to female ratio of 60:40 male to female is close to the usual reporting of 67%;33% but the demographic data clearly indicates that the population on were largely white 71%, urban residents 94% 

Of interest is the Comorbidity index at the 5+ level and the high rate of post traumatic seizures.   There also is as suspected a correlation to the pathophysiology of the TBI with SDH having the greatest correlation with the PTS.

It was not too surprising that those without PTS had a higher mortality rate.  Long term survival is also not correlated to seizures or post traumatic epilepsy.

The discussion comparing their findings to other studies as being similar is reassuring and sound and the discussion of pathophysiology of SDH and Seizures seems sound but the discussions in the following paragraphs line 240 - 256 are interesting but not needed to discuss this study and their findings.

Dear reviewer, thank you for your comments and great suggestion. We agree with your opinion. We summarized this section as suggested and removed redundancies. It reads as follows:

Traumatic Brain Injury Sub Types.

The type of TBI had a statistically significant effect on PTS. Traumatic SDH was associated with 57% of cases, while epidural hemorrhage, traumatic subarachnoid hemorrhage, concussion, traumatic cerebral edema, and diffuse traumatic brain injury were significantly fewer. The impact of traumatic SDH on PTS has been discussed previously in the literature. The epileptogenicity of SDH is secondary to the pronounced disruption of the blood brain barrier (BBB), facilitating the influx of various inflammatory mediators into the brain parenchyma. This neuroinflammatory milieu lowers the seizure threshold by promoting neuronal hyperexcitability [25-27]. Furthermore, the extravasated blood and its breakdown products trigger a series of cytotoxic and neuroinflammatory responses, collectively contributing to an environment conducive to seizures via GABAnergic inhibitory neurotransmission perturbance, gliosis, and neuronal loss [28-29].  The increased incidence of epileptogenicity following SDH can be seen in several recent studies, 24% in acute SDH and 11% in chronic SDH [21-23].  On the contrary, some studies also report lower incidences of PTS following SDH, with one study hypothesizing that this may be secondary to the mechanism of injury that causes SDH may increase mortality, thereby decreasing reported rates of PTS [24].”

The review of their limitations with the data from the NIS is helpful but the lack of EEG data is disconcerting, especially when it is now common practice for acute continuous EEG monitoring for those in ICU, at the highest risk of PTS.  Also noticeable is the lack of information regarding the use of prophylactic anticonvulsants in the first 7 days post TBI or what forms of neurosurgical intervention were needed or performed in response to the intracranial pathology which may be a key marker of severity/ complexity of the trauma.

Dear reviewer, thank you for the above comments and suggestions. Unfortunately, National Inpatient Sample data base does not report the information on medications and electroencephalogram. This is addressed in our limitations section: “One major limitation is that the NIS database doesn't report the clinical information such as physical exams, laboratory, imaging, EEG findings of the patients, and medications, so we could not elaborate and use these critical variables in our study to understand better the patients who developed PTS. One main limitation is that the current incidence of PTS is based on whether the institutions used routine continuous EEG to detect PTS.  However, some hospitals didn’t have continuous EEG or didn’t check for PTS.”

Finally their conclusions/recommendations over describe the NIS database as Robust - it is large in numbers but limited in other clinical ways and its population sample.  They also purport that early prophylaxis or AED use will lead to reduced PTS incidence ( a conclusion that may be correct but they have not proven) and I believe it is clear in the literature that distinction between early onset seizures and Late onset seizure post TBI need to be distinguished and addressed clinically in different ways.

Dear reviewer, thank you for your comments and suggestions. We have changed the designation of NIS from “robust” to “large in numbers” as follows: Our study was queried from NIS, a national healthcare database. While large in number, making this study strong for the epidemiological understanding of PTS to design future prospective studies, this database had a few limitations.

Regarding seizure prophylaxis, we have rewritten the conclusion as follows: Considering this high association, further research is warranted regarding early seizure prophylaxis and screening via techniques such as EEG in TBI management.

The distinction between early and late onset seizures following TBI has been added into the introduction with a new citation as follows: Seizures occurring within one week following TBI are considered provoked secondary to head injury; this is considered early PTS. After one week, the development of multiple seizures is considered post traumatic epilepsy and late PTS if only a single seizure occurs.

Reviewer 2 Report

Comments and Suggestions for Authors

Firstly, we would like to congratulate the authors on submitting their interesting research. We have a few comments about the manuscript:

1)    Please summarize the title. The study design type should still be included but authors can mention only that it is a retrospective study. Also, please substitute “;” with “ :”

2)    We suggest using English editing services to correct typos and improve the readability of the text.

3)    Introduction needs to be significantly improved, with relevant epidemiological data in percentages included, as well as diagnostic criteria and prognosis of PTS. Risk factor should be further characterized.

4)    Please include more data on TBI and PTS studies on the introduction, including experimental and clinical studies.

5)    PTS has been extensively over the decades. In the abstract and also in the last paragraph of the introduction, we suggest further specifying the research questions, goals and hypotheses, and conveying what is original about this article in particular.

6)    Please further explain the timeline of the study and when it began

7)    Please further explain the study protocol and how it was chosen.

8)    Please further explain inclusion and exclusion criteria for this study.

9)    Please add a separate statistical analysis section and provide detailed explanation of each statistical test chosen.

10) Was the power of the study calculated?

11) The discussion should include a detailed analysis of previous studies in the literature compared to the current results,

12) Please adjust the scale of the Graph in Figure 1

13) Please provide appropriate parameters of the logistic regression analysis on a table.

14) For all the analyses performed that were mentioned in the methods, there should be a proper display of parameters and results as well as analysis in the discussion. We suggest consulting a professional statistician for display of results.

15) Please follow STROBE guidelines for reporting of your data. You can attach a checklist with detailed explanations as supplementary data.

Comments on the Quality of English Language

Moderate editing required.

Author Response

Reviewer 2: Firstly, we would like to congratulate the authors on submitting their interesting research. We have a few comments about the manuscript:

  • Please summarize the title. The study design type should still be included but authors can mention only that it is a retrospective study. Also, please substitute “;” with “:”

Dear reviewer, thank you for your comments and suggestions. The title has been condensed and the semicolon replaced with a colon. It now reads as follows: “Development of Seizures Following Traumatic Brain Injury: A Retrospective Study.”

  • We suggest using English editing services to correct typos and improve the readability of the text.

Dear reviewer, thank you for your comments and suggestions. Our manuscript has now been edited and reviewed by a native English writer and readability should be improved.

  • Introduction needs to be significantly improved, with relevant epidemiological data in percentages included, as well as diagnostic criteria and prognosis of PTS. Risk factor should be further characterized.

Dear reviewer, thank you for your comments and suggestions. Epidemiological data in percentages is now reported in the introduction. These percentages were identified based on our database. Diagnostic criteria and prognosis of PTS have been added with additional resources. Risk factors has been further characterized. The abstract and introduction now read respectively as follows:

In this study, our goal was to discern and elucidate the incidence and risk factors implicated in the pathogenesis of PTS. We hypothesize that the development of PTS following TBI varies based on type and severity of TBI.”

“Traumatic brain injury (TBI) is a prevailing and economically burdensome issue within the United States, exacting a substantial toll through heightened morbidity and mortality. The Centers for Disease Control and Prevention (CDC) has underscored the magnitude of this concern, projecting over 64,000 TBI-related fatalities in 2020 and more than 223,000 TBI-associated hospitalizations in 2019 alone [1, 2]. The societal impact attributed to TBI is unequivocally reflected in the staggering estimated annual healthcare expenditure exceeding $40.6 billion [3]. Following TBI, patients are susceptible to various health impairments, psychosocial deficits, and neurological sequelae including the onset of post-traumatic seizures (PTS) [1, 3, 4]. Seizures occurring within one week following TBI are considered provoked secondary to head injury; this is considered early PTS. After one week, the development of multiple seizures is considered post traumatic epilepsy and late PTS if only a single seizure occurs [5]. The development of post-traumatic seizures has been seen to occur in 2% to over 50% of patients following head trauma; the large range of PTS occurrence attributed to injury severity and type of TBI [6]. The development of PTS worsens the prognosis of TBI patients by increasing ICU/hospital length of stay, ICU ventilation requirements, and results in poorer 24-month outcomes such as mortality and further development of post traumatic epilepsy [7]. PTE contributes to 3% to 6% of all new onset epilepsy and, therefore, warrants further investigation into risk factors such as patient demographics and comorbidities, medical complications, and TBI subtype [8].

Early PTS occurrence is attributed to the interplay of primary injury and excitotoxic milieus. Subsequent PTS occurrence is linked to secondary injury-associated changes such as neuroinflammation and blood-brain barrier perturbations [4]. Pre-clinical models have shown that glutamate signaling and GABA-A channels could play a significant role in early PTS occurrence. It is hypothesized that increased glutamate signaling occurs in response to decreased GABAnergic activity, likely secondary to microRNA regulation [6].  In addition to glutamate and GABA-A, reactive gliosis secondary to the upregulation of c-Jun N-terminal kinase (JNK) and the activating of the JNK signaling pathway by glutamate has been shown to increase seizure activity following TBI in animal models [10].  A multitude of risk factors for early PTS have been reported, including factors such as advanced age, alcohol misuse, low-energy trauma, distinct intracranial injury patterns, pre-existing medical comorbidities, and TBI severity [9, 11, 12]. However, discrepancies persist between these risk factors and the findings within the current research landscape [13, 14].

Comprehending and characterizing the incidence and risk factors for PTS is paramount, as it raises healthcare demands, increases economic burdens, and reduces functional outcomes [15, 16]. In this study, we used a large national database and explored all patients with a principal admission for TBI. Research is scarce regarding the incidence of PTS following TBI on a national scale in the United States. Our intention in this review is to discern the incidence of in-hospital PTS and its associated risk factors. Our goal is to identify areas where preventive measures can be enacted to reduce PTS occurrence in the future. We hypothesize that the incident of seizure following TBI varies based on type and severity of brain injury.”

  • Please include more data on TBI and PTS studies on the introduction, including experimental and clinical studies.

Dear reviewer, thank you for your comments and suggestions. As response to number three, three additional studies have been included in the introduction to further address epidemiological data, diagnostic criteria, and prognosis of PTS.

  • PTS has been extensively over the decades. In the abstract and also in the last paragraph of the introduction, we suggest further specifying the research questions, goals and hypotheses, and conveying what is original about this article in particular.

Dear reviewer, thank you for your comments and suggestions. Research questions and goals are now clarified in the introduction. It now reads as:

Comprehending and characterizing the incidence and risk factors for PTS is paramount, as it raises healthcare demands, increases economic burdens, and reduces functional outcomes [12, 13]. In this study, we used a large national database and explored all patients with a principal admission for TBI. Research is scarce regarding the incidence of PTS following TBI on a national scale in the United States. Our intention in this review is to discern the incidence of in-hospital PTS and its associated risk factors. Our goal is to identify areas where preventive measures can be enacted to reduce PTS occurrence in the future. We hypothesize that the incident of seizure following TBI varies based on type and severity of brain injury.”

  • Please further explain the timeline of the study and when it began

Dear reviewer, thank you for your comments. The timeline of the study is included in the methods section as follows: “The National Inpatient Sample (NIS) database from the Healthcare Cost and Utilization Project (HCUP) was queried for all primary cases with a principal diagnosis of traumatic brain injury (TBI) from January 2016 to December 2020.”

  • Please further explain the study protocol and how it was chosen.

Dear reviewer, the study protocol was based off the interest between TBI and posttraumatic seizures. Methodology was like many other NIS studies cited in our resource section.

  • Please further explain inclusion and exclusion criteria for this study.

Dear reviewer, thank you for your comments. Inclusion and exclusion criteria for this study can be seen in the following paragraph that is within the methods section:

“Using the International Classification of Diseases, Tenth Revision (ICD-10) codes, we extracted cases with the ICD-10 code S06 family for intracranial injury (S06.0-S06.9) and the ICD-10 code R56.1 for post-traumatic seizures (PTS). After properly excluding cases with incomplete and missing variables, the sample was divided into two groups based on the presence or absence of seizures secondary to TBI during the inpatient stay. Inclusion criteria included any patient with ICD-10 codes for TBI and post-traumatic seizures. Exclusion criteria included any patient with incomplete data.”

  • Please add a separate statistical analysis section and provide detailed explanation of each statistical test chosen.

Dear reviewer, thank you for your suggestion. We have added a separate statistical analysis section as follows:

Categorical results are reported as counts with column percentages. Continuous data are reported as means with standard deviations (SD); standard errors (SE) are provided where appropriate. A comparison of normally distributed data was performed using independent sample t-tests. For non-normally distributed data, the Wilcoxon rank-sum test was applied. Categorical variables were assessed using Fisher’s Exact Test or Pearson's Chi-Square test with Kendall's Tau-b. Where appropriate, residuals were evaluated to ensure normal distribution, and no issues of multicollinearity were observed. Regression results are reported as adjusted odds ratios (OR) with 95% confidence intervals (95% CI). For the analysis of days to the primary outcome (in-hospital mortality), Kaplan-Meier survival estimates and log-rank tests were used to compare TBI patients with and without PTS. 

Statistical analyses were performed using R version 4.2.0 (R Core Team, 2022), a statistical computing environment developed by the R Foundation for Statistical Computing, Vienna, Austria. The significance level was set at p < 0.05. As the data extracted from the NIS are not linked to any patient identifiers, this study was deemed exempt from full IRB review by the University of Texas Health Science Center at San Antonio (HSC20150408N).”

  • Was the power of the study calculated?

Dear reviewer, thank you for your question. Yes, the power was calculated. Usually for large studies like this the power is extremely high, we have a calculated power of 0.99.

  • The discussion should include a detailed analysis of previous studies in the literature compared to the current results

Dear reviewer, thank you for your comments and suggestion. We have included multiple previous studies in our discussion section and a detailed analysis comparing the current results can be seen.

  • Please adjust the scale of the Graph in Figure 1.

Dear reviewer, thank you for your suggestion. The adjusted graph is attached below.

  • Please provide appropriate parameters of the logistic regression analysis on a table.

Dear reviewer, thank you for your suggestion. Please see the comments below for parameters of the logistic regression analysis.

“Table 2. Logistic regression analysis of the odds of developing post-traumatic seizures (PTS) following a primary traumatic brain injury (TBI), stratified by demographic variables. The model includes adjustments for age, gender, race, and comorbidity burden as measured by the Elixhauser Comorbidity Index. Odds ratios (OR) with 95% confidence intervals (CI) are presented, with statistical significance indicated at p < 0.05.”

  • For all the analyses performed that were mentioned in the methods, there should be a proper display of parameters and results as well as analysis in the discussion. We suggest consulting a professional statistician for display of results.

Dear reviewer, thank you for your suggestion. All parameters are mentioned in the methods as well as the figure legends.

  • Please follow STROBE guidelines for reporting of your data. You can attach a checklist with detailed explanations as supplementary data.

Dear reviewer, the STROBE checklist has been uploaded under the non-published material section.
